# Chemoresistive Sensors for Cellular Type Discrimination Based on Their Exhalations

**DOI:** 10.3390/nano12071111

**Published:** 2022-03-28

**Authors:** Michele Astolfi, Giorgio Rispoli, Mascia Benedusi, Giulia Zonta, Nicolò Landini, Giuseppe Valacchi, Cesare Malagù

**Affiliations:** 1Department of Physics and Earth Sciences, University of Ferrara, 44122 Ferrara, Italy; giulia.zonta@unife.it (G.Z.); nicolo.landini@unife.it (N.L.); malagu@fe.infn.it (C.M.); 2SCENT S.r.l., Via Quadrifoglio 11, 44124 Ferrara, Italy; 3Department of Neuroscience and Rehabilitation, University of Ferrara, 44121 Ferrara, Italy; mascia.benedusi@unife.it; 4Department of Environmental Science and Prevention, University of Ferrara, 44121 Ferrara, Italy; giuseppe.valacchi@unife.it; 5Plants for Human Health Institute, NC State University, Kannapolis, NC 28081, USA; 6Department of Food and Nutrition, Kyung Hee University, Seoul 130-701, Korea

**Keywords:** chemoresistivity, nanostructures, sensors, cells, tumor, VOCs

## Abstract

The detection of volatile organic compounds (VOCs) exhaled by human body fluids is a recent and promising method to reveal tumor formations. In this feasibility study, a patented device, based on nanostructured chemoresistive gas sensors, was employed to explore the gaseous exhalations of tumoral, immortalized, and healthy cell lines, with the aim of distinguishing their VOC patterns. The analysis of the device output to the cell VOCs, emanated at different incubation times and initial plating concentrations, was performed to evaluate the device suitability to identify the cell types and to monitor their growth. The sensors ST25 (based on tin and titanium oxides), STN (based on tin, titanium, and niobium oxides), and TiTaV (based on titanium, tantalum and vanadium oxides) used here, gave progressively increasing responses upon the cell density increase and incubation time; the sensor W11 (based on tungsten oxide) gave instead unreliable responses to all cell lines. All sensors (except for W11) gave large and consistent responses to RKO and HEK293 cells, while they were less responsive to CHO, A549, and CACO-2 ones. The encouraging results presented here, although preliminary, foresee the development of sensor arrays capable of identifying tumor presence and its type.

## 1. Introduction

Tumor detection and monitoring are fundamental topics in medicine worldwide. According to American Cancer Society statistics, one out of three people will be diagnosed with cancer in their lifetime [1,2]. For this reason, the development of screening and monitoring methods, able to detect tumors at the earliest possible stage, is of paramount importance for cancer treatment, maximizing the chance to eradicate them. In order to carry out widespread screening protocols (possibly all over the entire population of a country), it is crucial to maximize the effectiveness of these methods, first by minimizing their cost and invasiveness.

Tumor gaseous biomarkers are volatile organic compounds (VOCs) produced by cancer cells mainly as a result of cell metabolic process alterations [3] and cell plasma membrane peroxidation [4]. The identification of VOCs exhaled by human body fluids (as urine [5,6,7], breath [8,9,10,11,12,13,14,15], blood [16,17], and feces [18,19]), exploiting a low invasive technique, is crucial to detect tumor formations, minimizing the collateral effects and injuries to the patients [20,21,22,23,24]. In previous studies of this research team, patented devices based on nanostructured chemoresistive gas sensors technology, developed in Laboratorio Sensori of the University of Ferrara, have been designed and manufactured to analyze feces exhalation for colorectal cancer (CRC) screening, blood exhalations for tumor post-surgical and post-treatment monitoring, and CRC tissue exhalations [25,26,27,28,29,30,31,32,33,34,35]. Recently, the study of cultured cells exhalations, due to their simplicity and reproducibility, has become popular to identify VOCs from different tumor types [36,37,38,39]. On this ground, this paper is focused on the discrimination of six different cell culture types (A549, CACO-2, CHO, HEK293, fibroblasts, and RKO) on the basis of their VOC pattern. The cell VOCs were detected by an innovative portable device based on an array of nanostructured thick film sensors. The aim of this study is the preliminary evaluation of the device suitability to identify the cell type, and at the same time to monitor the cell health status and growth as a function of their initial plating concentration and incubation time. These cell lines, being tumoral, immortalized, and healthy (used as controls), were chosen to have a wide array of sample types to challenge the sensors. The ultimate goal of this basic research is the development of a fingerprint cellular database on the basis of cell VOC emissions, and at the same time to find out the most suitable sensor array to identify and classify human tumors. In this context, it is therefore unnecessary to develop sensor arrays able to single out the molecular composition of cellular VOCs (that would require a large number of sensors), but to gather instead a limited set of sensors able to univocally identify the cell lines, regardless of how selective each sensor is.

## 2. Materials and Methods

### 2.1. The Sensor Working Principle and the Sol-Gel Technique

Each sensor comprised a nanostructured-sensitive film formed by nanograins of metal-oxide semiconductor material (MOX), entirely designed, produced, and assembled in the “Laboratorio Sensori” of the University of Ferrara, Italy. The working principle of these sensors is based on the chemoresistivity, i.e., the capability of varying the film conductivity as a consequence of the chemical reactions occurring between the surface of the grains and the gaseous molecules of the surrounding atmosphere. The chemoresistivity process requires the thermal activation of the sensor that consists in maximizing the number of electrons (in conduction band) able to overcome the potential barrier (band bending, BB in Figure 1) between grains, and in the thermal ionization of the sample gases. Therefore, it is necessary to heat the sensor to an appropriate working temperature. When the MOX sensor is placed in dry air (i.e., in the reference atmosphere), the surrounding oxygen ions and/or molecules are adsorbed on the grain surface (Figure 1a, left), and this occurs on the bulk grains as well, because of the porous structure of the sensor thick film. The adsorbed oxygen ions act as acceptor surface states because of their electronegativity; therefore, they reduce the sensing film conductivity by trapping conduction band electrons to the grain surface (Figure 1a, left). When the oxygen coverage ratio does not change in time, i.e., when the number of adsorbed oxygens per unit of time is equal to the desorbed ones, an equilibrium state is achieved, and film conductivity becomes constant. Moreover, for an n-type sensor, the oxygen adsorption enlarges the grain “depleted shell” (i.e., the grain region lacking free charges; Figure 1a, pink ring), emphasizing the “band bending” effect [40,41] and consequently decreasing the sensor conductivity. At this point, if a test-reducing gas as CO is applied to the sensor film, it reacts with the previously adsorbed oxygen ions to form CO_2_ molecules that are quickly released into the atmosphere (Figure 1a, right). This frees the grain surface states and allows the electrons to return to the conduction band as well, increasing the film conductivity; once the rate of this reaction has reached equilibrium, film conductivity is again constant. The substitution of the test gas with the dry air (reference atmosphere) desorbs the gaseous molecules entrapped on the sensor surface, allowing for the complete recovery of the sensor to its initial state (i.e., prior to the test gaseous exposure). This returns the sensor conductivity to its starting (reference) value, because of the full reversibility of the chemical reactions occurring on the grain surface.

To optimize the performances of a chemoresistive MOX gas sensor, it is primarily necessary to increase the surface–volume ratio of its sensitive porous material as much as possible, to maximize its interaction with the surrounding gases. With this aim, the sol-gel chemical process was employed to synthesize networks of MOX nanometric spherical grains, precisely interconnected [42,43,44,45]. The sol–gel process consists of the following main steps:The hydrolysis of the precursor (as silicon or metal alkoxides) in water or alcohol solution (named “sol”, as solution, in Figure 1b); usually some other chemicals (as acids, etc.) are added to the solution to promote the sol process that foresee the formation of a colloidal suspension.The condensation of the MOX molecules (“Aggregation”, Figure 1b), occurs when the water or alcohol are removed and the MOX bridging takes place. This forms a colloidal and viscous network, but in a liquid phase (“GEL”, Figure 1b); the alkoxide precursor and pH of the solution are the two parameters that mainly influence the colloidal particles size and cross-linking.The aging process (up to a couple of days), during which several changes in the gel structure and properties occur as polycondensation, lead to an increase in thickness of the colloidal particles and to the reduction of their porosity.The last step is the drying and calcination: the GEL is dried first (at about 100 °C), and then calcinated at higher temperatures (at about 400–800 °C) to completely remove the residuals of solvents and of other chemical additives. In this process, the temperature and the relative humidity greatly influence the quality of the final MOX powder.

### 2.2. The Sensor Assembly

The MOX nanopowder, synthesized through the sol–gel technique [41,42,43], was then converted in a printable viscous paste following the addition of a small amount of organic vehicles (as α-tertineol) and a glass frit (a mixture of glassy silicon oxides). The final MOX nanostructured paste was distributed on an alumina substrate between the two gold contacts by means of a screen-printing machine (mod. C920, Aurel s.p.a., Forlì-Cesena, Italy). The printed substrate was dried first (at about 100 °C) and then subjected to a firing thermal treatment (up to 850 °C) to fabricate a square film (250 × 250 µm, thickness 25–30 µm) of MOX nanograins of size ranging between 50 and 200 nm (Figure 2), which was finally welded in the sensor assembly. The latter was composed of three components (Figure 3a–c): a substrate, a sensitive material film, and a heater. The substrate was made of sintered alumina that provided a stiff and insulating layer, hosting two interdigitated gold contacts (printed by means of a lithographic process), necessary to connect the sensor to the signal transduction circuit. The heater consisted of a platinum coil placed below the substrate aimed to heat up the sensor to the proper working temperature by precisely controlling the current flowing through the coil. The three components were welded together by thermo-compression bonding, and the sensor gold contacts and the heater were connected to a four-pin TO-39 socket, such that the completed sensor could be simply connected to the transduction and heating electronic circuits and could be easily replaced (Figure 3a,b).

### 2.3. The Sensor Array, the Response, and the Device

The four MOX sensors hosted in SCENT B1, listed below, were selected on the basis of previous laboratory tests carried out on pure standard gases, on VOCs exhaled by blood, feces, biopsies, and preliminary tests on cultured cells [25,26,27,28,29,30,31,32,33,34]: ST25, based on tin oxides and titanium (25%) and gold nanoparticles (1%);W11, based on tungsten oxide;STN, based on tin, titanium, and niobium oxides;TiTaV, based on titanium, tantalum and vanadium oxides.

All sensors were thermally treated with a firing temperature of 650 °C and were heated to the working temperature of 450 °C by the platinum coil [31,32,33,34]. Each sensor was connected to the SCENT B1 electronic system via four pins: two were wired to the sensor heating system and two to the signal transduction circuit (Figure 3). The latter consisted of an inverting operational amplifier that converted the sensor conductivity changes to a voltage difference (Figure 3d), i.e., the amplifier output voltage $V_{o}$ was directly proportional to the supply voltage $V_{i}$ and to the sensing film conductivity [31,32]. The output voltage signals were sampled and digitized by an on-board A/D converter; custom software acquired and plotted them in real time. To have a dimensionless physical quantity related to the sensor interaction with gaseous molecules, each sensor response was computed as:(1)$$R=\frac{V_{gas}}{V_{air}}$$
where $V_{gas}$ and $V_{air}$ are the steady state sensor signal in the presence of the sample gas and in the presence of the environmental air only, respectively (Equation (1)).

In order to assess the reliability of the sensor responses on the cell samples presented in the following, four different STN sensors were tested, as an example, in both wet (RH = 30%; Figure 4a) and dry (RH = 0%; Figure 4b) atmospheres, under decreasing CO concentrations, from 40 to 0 ppm. The sensor responses appear repeatable, showing a relative difference no larger than ~13%.

The patented device [46] employed here, SCENT B1 (described in detail in [31,32,33]), was mainly composed of two parts: a pneumatic and an electronic system (Figure 5). The former conveyed to the four different sensors (by means of an electric pump) the environmental air stabilized in humidity by a carbon filter and sterilized by a 0.2 µm nylon filter (changed before each experiment); the humidity stabilization and the air pollutant removal were crucial to have a steady response to the environmental air (baseline) [31,32,33]. The electronic system consisted of dedicated electronic boards to provide the device power supplies, to control the sensor heating, and to digitalize and acquire the sensor signals (Figure 2).

### 2.4. Cell Types and Sample Preparation

The cell types employed here, purchased from the American Type Culture Collection (ATCC^®®^ CCL-61™, Manassas, VA, USA), but the human skin fibroblasts were:A549, explant culture of lung carcinomatous tissue;CACO-2, human colorectal adenocarcinoma cell line;CHO, epithelial cell line derived from the ovary of the Chinese hamster;HEK293, specific immortalized cell line originally derived from human embryonic kidney;RKO, human colorectal cancer cells.

Human skin fibroblasts were obtained during routine health checks or by donations, isolated by a 3 mm skin punch biopsy and used between the third and fifth passage in vitro [47]. Since the cells were kept and grown in media rich in organic compounds as amino acids and vitamins, it is expected that the media exhaled some compounds, interfering with the cell measurements, as it was found (see Results). DMEM (Dulbecco’s Modified Eagle’s Medium, containing 10% FBS, 1% of L-glutamine and 1% of penicillin/streptomycin, all from Lonza, Milan, Italy) and RPMI (Roswell Park Memorial Institute; Lonza, Milan, Italy), supplemented with various FBS (Fetal Bovine Serum; Euroclone, Milan, Italy) fractions (5% and 10%), were tested to explore their VOCs for present and future applications. All cells were grown and maintained in DMEM: cell responses $R_{c}$ (calculated according to Equation (1)) were therefore scaled to the DMEM response $R_{m}$ (that was routinely evaluated before and after each round of cell measurements). To remove the media contribution to each cell sensor response ($R_{c})$, it was thought to normalize the latter as follows:(2)$$R_{N}=\frac{R_{c}}{R_{m}}$$
and $R_{N}$ was reported in each plot abscissae.

Since the chemoresistive sensors are highly sensitive to the gas temperature, the dishes (closed with their cover) taken out from the incubator were quickly cooled to room temperature on a copper plate. Room temperature was monitored and kept approximately constant at 22.0 ± 1.5 °C.

To maximize the sample exhaling surface, in order to improve the signal-to-noise ratio, three petri dishes were simultaneously measured by placing them in the sample chamber stacked in an holding tray (Figure 6). Each round of experiments consisted of the measurement of DMEM only, the cell samples at various concentrations after 24, 48 and 72 h from plating, and again DMEM only. Of the nine cell dishes at each concentration, three were withdrawn from the incubator after 24 h of plating, measured and then discarded, three were measured after 48 h, and the last three after 72 h. Therefore, it was necessary to prepare eighteen petri dishes with DMEM and nine petri dishes for each cell concentration chosen, namely 8 × 10^4^, 2.5 × 10^5^, 5 × 10^5^, 1 × 10^6^ cells per dish (3.5 cm of diameter); all dishes were kept in the same incubator at 37 °C in a humidified atmosphere containing 5% of CO_2_.

## 3. Results and Discussion

### 3.1. Sensor Responses to the Culture Media

Since the cells were kept and grown in media rich in organic compounds, it was necessary to measure $R_{m}$ to assess its contribution to $R_{c}$. Two media, DMEM and RPMI supplemented or not with various concentration of FBS, were measured at 37 °C or at room temperature 24 or 72 h after withdrawal from the incubator, in order to mimic as much as possible the cell experimental conditions, employed in the following.

The $R_{m}$ amplitude of DMEM and RPMI were similar and not affected by the presence of FBS nor by the dwell time in the incubator, but at 37 °C, they were significantly larger than at room temperature. These results are predictable, because the culture media are not expected to change their VOC emission with time if they were properly handled (i.e., no organic contaminant was grown in the media). However, the sensor response to H_2_O at 37 °C (Figure 7a) showed the non-negligible response to the cell culture media. Therefore, to simplify the experimental procedures, it was decided to perform all the cell measurements at room temperature, and to normalize $R_{c}$ to $R_{m}$ according to Equation (2).

### 3.2. Sensor Responses to RKO Cell Line

The largest sensor responses were attained with human colorectal cancer cells (RKO; Figure 8). All sensors gave progressively increasing responses (but W11) upon the cell density increase and withdrawal time from incubator; ST25 and STN sensors gave the clearest responses (Figure 8). This simply means that the sensor signals increased along with the cell number, no matter if this number was attained by the initial seeding or by the cell growth.

Cells seeded at the three concentrations grew readily from one day to the next such that the ones at 250 K/dish were almost confluent after 72 h (Figure 9, orange background), the 500 K ones were fully confluent after 48 h (Figure 9, green) and the 1 M ones started to die at 48 h (Figure 9, green), probably for a significant run down of medium nutrients. This was readily detected by the sensors, whose responses to the 1 M cells/dish density at 72 h (Figure 8), although larger than the one at 24 h, were smaller than the one at the 500 and 250 K concentrations (at 72 h).

Figure 9 (orange background) shows, at high magnification, the cells plated at 1M after 72 h of plating that had become spherical and granular, a clear indication that they were detaching from the dish bottom and dying. A further reason of the small response at 1 M cells/dish concentration could be a reduced cell growth rate and/or activity due to contact inhibition, and consequently a decrease in VOC exhalation. However, this possibility is excluded, since cancerous cells continue proliferation and growth when they contact each other, leading to uncontrolled cell propagation and solid tumor formation [48,49].

### 3.3. Sensor Responses to HEK-293 Cell Line

The next largest responses were attained by using HEK-293, a specific immortalized cell line originally derived from human embryonic kidney. Again, all sensors gave progressively increasing responses (but W11) upon the cell density increase and withdrawal time from incubator (Figure 10).

However, all sensors poorly discriminated among the 80, 250, and 500 K cells/dish plated at 24 h, except for the 1 M one. To check that the responses to the first three cell concentrations were small, they were averaged (n = 3, standard error of ±0.02) and are shown in Figure 10; regardless, a normalized response below 1.1 can be considered negligible. Again, the 1 M cell concentration after 72 h from plating gave a response that was smaller than the one at 48 h, because the cells were confluent. This decrease was however less marked than the RKO one, probably because the number of deteriorating cells was smaller in the former case than in the latter one (Figure 11).

### 3.4. Sensor Responses to CHO and A549 Cell Lines

Sensor responses to CHO cells, an epithelial line derived from the ovary of the Chinese hamster, and A549 cells, an explant culture of lung carcinomatous tissue, were negligible after 24 h of plating, regardless of the plating concentration. In particular, the A549 responses were smaller than one, i.e., they were smaller than the sensor response to culture medium only. Both these cell types grew quite fast, and at a concentration of 500 K, cells/dish were already confluent after 48 h of plating (Figure 12, insert); therefore, these cells were not plated at 1 M cells/dish. As usual, ST25, STN, and TiTaV gave progressively increasing responses (but W11) upon the cell density increase and withdrawal time from incubator (Figure 12). The rapid cell confluence reached by these two cell lines gave similar response amplitudes at 72 h regardless of the plating concentration. The A549 cells exhalations were detected only at confluence (i.e., after 48 h of plating) and only by the ST25 and TiTaV sensors.

### 3.5. Sensor Responses to CACO-2 and Fibroblast Cell Lines

Lastly, the cell lines CACO-2 and human skin fibroblasts gave negligible responses (Figure 13). This was expected for the fibroblasts since they are not tumoral cells and their normal metabolism is expected to emit VOCs at a lower rate in respect to the tumorigenic ones. The CACO-2, derived from human colorectal adenocarcinoma, could also be expected to emit a lower amount of VOCs in respect to more invasive cancer cell lines, since it is a poorly aggressive tumor, and it has been widely adopted as a model of the intestinal epithelial barrier to study the mechanisms implicated in early-stage cancer progression, and to test the radiation therapeutic efficacy [50]. However, it is not expected, even after 72 h of plating, that the CACO-2 will become differentiated polarized epithelial cells (a stage that is expected to emit low amounts of VOCs), since this happens about after 20 days of plating [51].

### 3.6. Sensor Responses as Fingerprint for Cell Typing

In summary, it is possible that the VOCs emitted by CHO, A549, and CACO-2 cell lines, although they are expected to be quite sustained since these cells are fast growing and therefore have a high metabolism, are poorly detected by the sensors. If this is the case, it may become advantageous, since the sensors here employed, giving different response patterns with different cancer cells, could be used in the future not only to detect them (hopefully to an early stage), but also to identify their type. To further explore this idea, as shown in Figure 14, the responses of the sensors to different cell lines at the same plating concentration and withdrawal time from the incubator, when they were close to confluence (i.e., 250 K cells/dish after 48 h of plating), are compared.

The same sensor gave different response amplitudes to different cell lines (Figure 14a), or different sensors gave different response amplitudes to the same cell line (Figure 14b). Tumoral cells usually have higher metabolism in respect to healthy cells, releasing many substances, in particular VOCs, that acidify the extracellular milieu [52]. Since the sensors employed here are made by n-type MOX materials, their responses to tumoral cells are higher than the ones to healthy cells because they interact with the tumoral reducing gases (see Material and Methods). However, it is expected that different tumoral cells produces different VOCs spectra, which are differently sensed by the different MOX films. This indicates that a particular cell type could be identified by properly combining the response to its VOCs by different sensors. Operatively, a matrix representing the response of one sensor (for instance the one that gave the smallest response, indicted with red numbers in Table 1) to a particular cell line and the ratios between the response of the other two sensors and the former one to the same cell line (Table 1; W11 was excluded in this computation since it gave unreliable responses) can be constructed. The resulting matrix can be tentatively interpreted as a “fingerprint response” of the set of the three sensors that would identify the cell line. The outcome is encouraging, because the matrix univocally identifies the corresponding six cell lines. For this strategy to be reliable, it is necessary that, as the cell number increases (but avoiding the cell confluence), the responses of all the sensors change by the same amount, such that the ratios are independent of the cell number, and this appears to (roughly) be the case (Figure 8, Figure 10 and Figure 12). However, to construct reliable matrices as the one in Table 1, it is necessary to repeat many times the above experiments and report the averaged responses with their standard errors. This however requires a (costly) large number of cells, and regardless, the good reproducibility of these experiments suffices to draw consistent deductions.

## 4. Conclusions

The sensors ST25, STN, and TiTaV used in this work gave progressively increasing responses upon increasing the cell number, no matter if it was attained by the initial seeding concentration or by the cell growth. The sensor W11 gave instead unreliable responses to all the cell lines here employed, and therefore, they were discarded in the final analysis. The sensors gave large responses to the RKO and HEK-293 cell lines, while they were progressively less responsive to the CHO, A549, and CACO-2 cell lines. This suggests that a particular cell type could be identified by properly combining the response to its VOCs via a different sensor, allowing for the future development of sensor sets that are able to identify the presence of a tumor and even its type. The novelty of this approach, in respect to the current methods in development for tumor identification, is the employment of a portable, compact, and fast (the outcome is attained within 10–15 min) device, not requiring expensive consumables or complex and expensive analysis of the samples (that may require days), and that can be operated by untrained people.

## 5. Patents

Malagù, C.; Gherardi, S.; Zonta, G.; Landini, N.; Giberti, A.; Fabbri, B.; Gaiardo, A.; Anania, G.; Rispoli, G.; Scagliarini, L. Combinazione di materiali semiconduttori nanoparticolati per uso nel distinguere cellule normali da cellule tumorali (2015), National #:102015000057717 [40].

## Figures and Tables

**Figure 1 nanomaterials-12-01111-f001:**
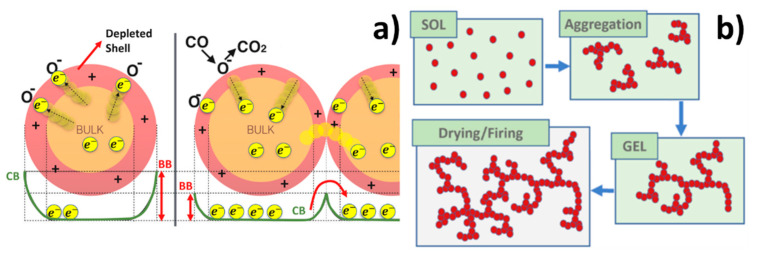
Synthesis and working principle of the MOX nanostructured thick film. (**a**) The four main steps of the sol-gel process producing interconnected networks of nanometric grains. (**b**) N-type sensor detection process to a reducing gas (CO). CB and BB are the conduction band and the band banding, respectively; yellow points represent electrons, orange circles the grain bulk, and the pink ring is the grain depletion region.

**Figure 2 nanomaterials-12-01111-f002:**
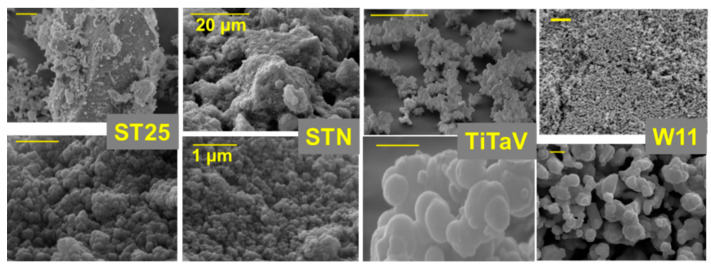
SEM images of nanostructured films. The ST25, STN, TiTaV, and W11 sensor thick films (from **left** to **right**) are displayed at low (**upper panels**) and high (**lower panels**) magnifications.

**Figure 3 nanomaterials-12-01111-f003:**
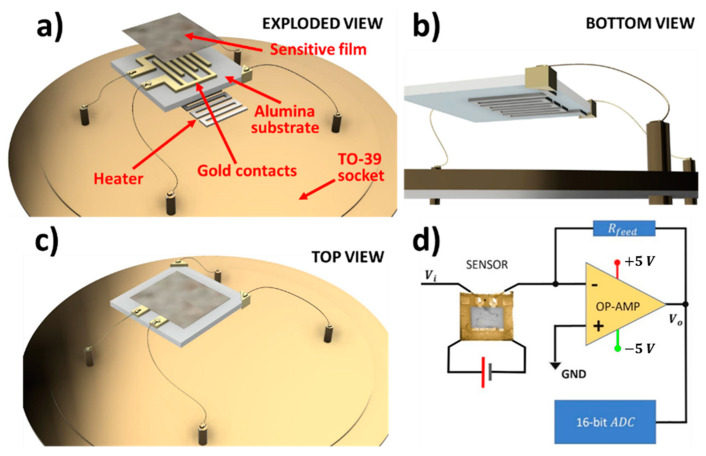
The sensor, its parts and its connection to the electronic circuit. (**a**) Exploded views of the sensor assembly; (**b**) bottom view; (**c**) top view; (**d**) outline of one sensor circuitry: microphotograph of a sensor (2.5 × 2.5 mm) connected in between the voltage source $V_{i}$ and the operational amplifier in an inverting configuration; the sensor heater contacts are connected to an adjustable voltage source; the output signal $V_{o}$ is digitized by a 16 bit analog-to-digital converter and sent to an external host computer via a USB serial port.

**Figure 4 nanomaterials-12-01111-f004:**
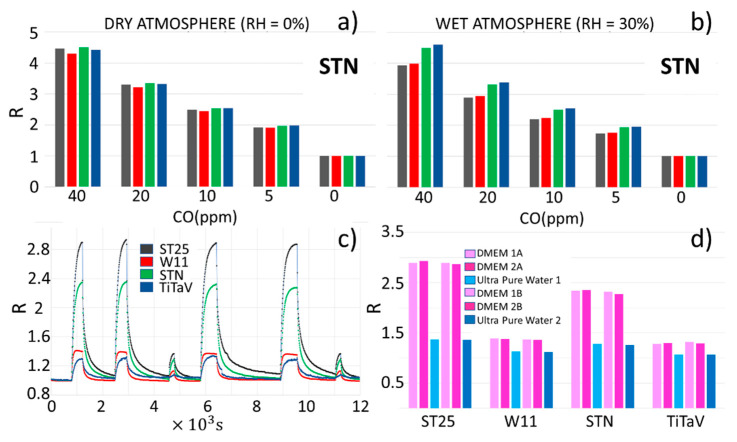
Sensor reproducibility and measure repeatability. (**a**) Response amplitudes (R) of four different STN sensors exposed to decreasing CO concentrations (from 40 to 0 ppm) in dry atmosphere. (**b**) Same as (**a**), but in 30% relative humidity; (**c**) plot of R vs. time of the sensors equipping the SCENT B1 (ST25, STN, TiTaV, and W11), exposed repetitively to different batches of DMEM and pure water to assess the measures’ repeatability; (**d**) bar graph of R for each sensor to the samples of (**c**).

**Figure 5 nanomaterials-12-01111-f005:**
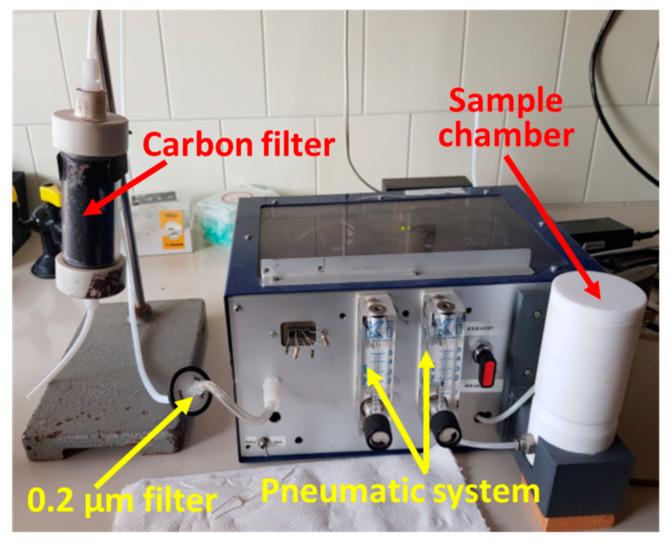
The device and its parts. The case is 30 × 15 × 21 cm^3^ (WxHxD).

**Figure 6 nanomaterials-12-01111-f006:**
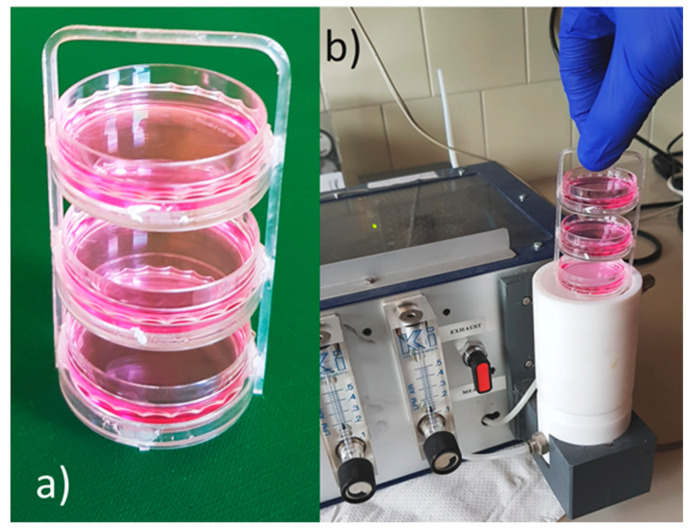
Sample insertion into the device. (**a**) Three petri dishes containing the cells, or the culture medium only, positioned in the holding tray; (**b**) the tray holding the three petri dishes being placed in the SCENT B1 sample chamber.

**Figure 7 nanomaterials-12-01111-f007:**
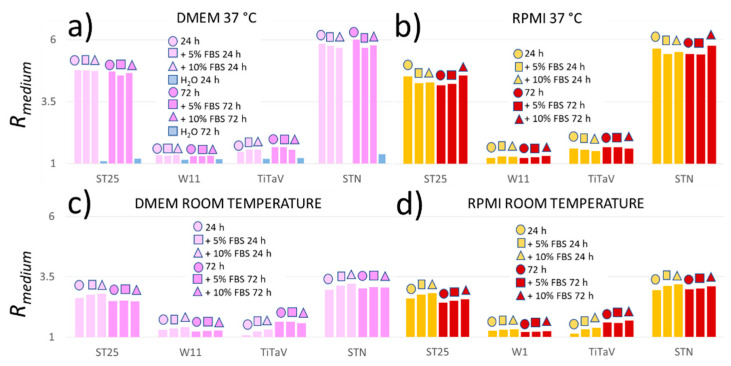
Sensor responses to different culture media. (**a**) $R_{m}$ amplitude to DMEM only, DMEM supplemented with 5% or 10% FBS, and ultrapure water measured 24 or 72 h after withdrawal from the incubator at 37 °C; (**b**) $R_{m}$ amplitude to RPMI only, RPMI supplemented with 5% or 10% FBS, at 37 °C 24 or 72 h after withdrawal from the incubator; (**c**) same responses of (**a**) but at room temperature; (**d**) same response of (**b**) but at room temperature.

**Figure 8 nanomaterials-12-01111-f008:**
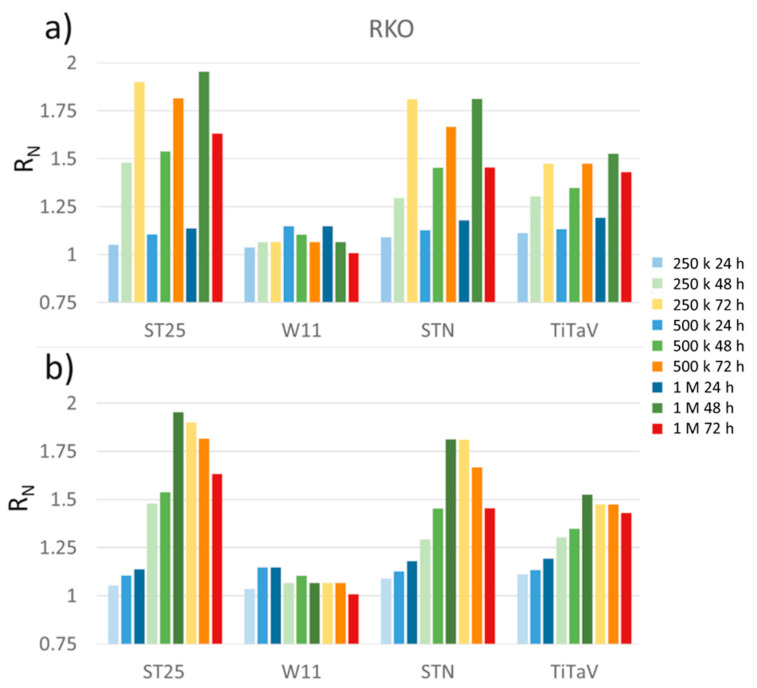
Sensor responses to RKO cell line. Cells plated at 250 K (light colors), 500 K (medium dark colors), and 1 M (dark colors) cells/dish, measured after being withdrawn from the incubator at 24 h (blue), 48 h (green), and 72 h (orange); (**a**) cells sorted for the same concentration at different times of incubation; (**b**) cells sorted for the same time of incubation but at different concentrations.

**Figure 9 nanomaterials-12-01111-f009:**
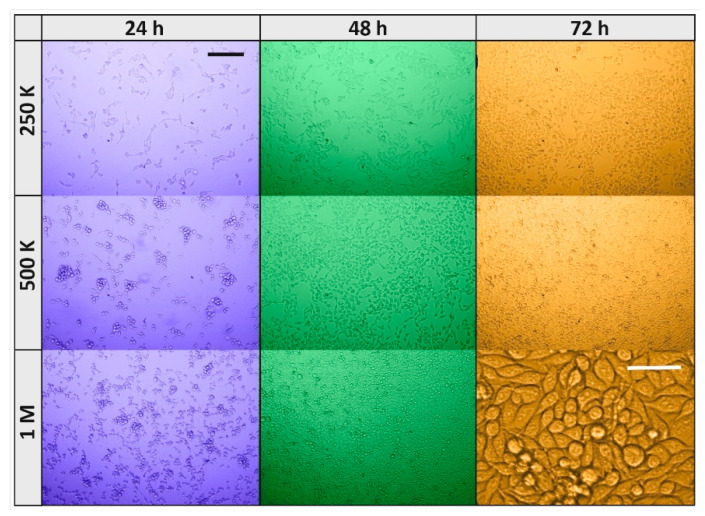
Microphotograph of RKO cells plated at various times of withdrawal from the incubator and at different densities. Cells plated at 250 K, 500 K and 1 M cells/dish withdrawn from the incubator after 24 h (blue background), 48 h (green), and 72 h (orange); black scale bar is 200, and it applies to all microphotographs, but the white scale bar (50 µm) applies to its microphotograph only.

**Figure 10 nanomaterials-12-01111-f010:**
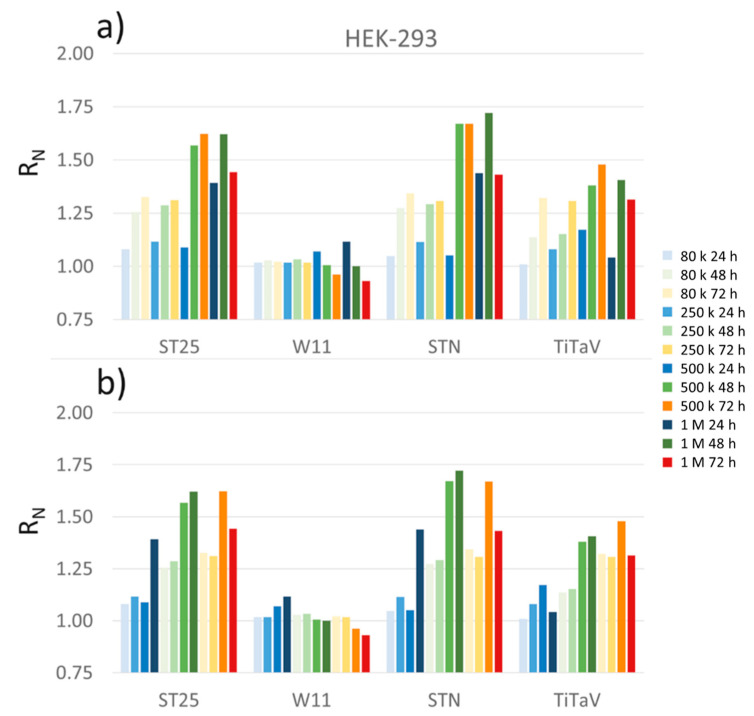
Sensor responses to HEK-293 cell line. Cells plated at 80 K (lightest colors), 250 K (light colors), 500 K (medium dark colors), and 1 M (darkest colors) cells/dish, measured after withdrawn from the incubator at 24 h (blue), 48 h (green) and 72 h (orange); (**a**) cells sorted for the same concentration at different times of incubation; (**b**) cells sorted for the same time of incubation but at different concentrations.

**Figure 11 nanomaterials-12-01111-f011:**
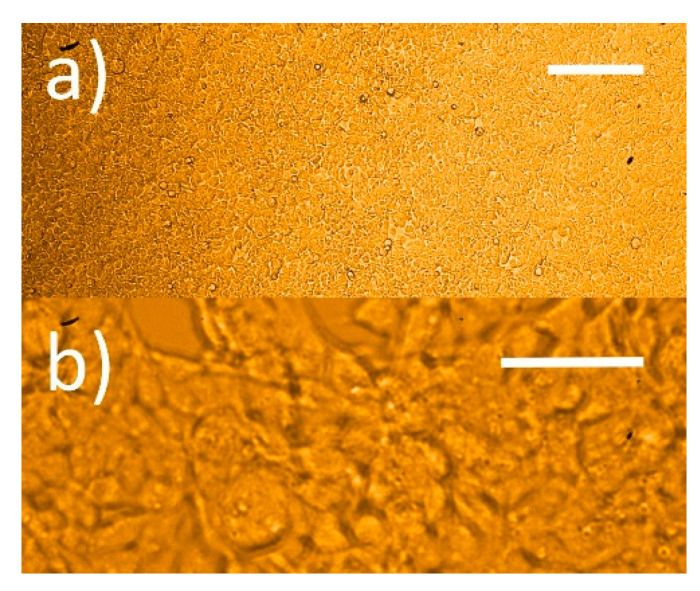
Microphotograph of confluent HEK-293. (**a**) Cells withdrawn from the incubator at 72 h initially plated at 1 M cells/dish, scale bar is 200 µm; (**b**) cells in a) at higher magnification, scale bar is 50 µm.

**Figure 12 nanomaterials-12-01111-f012:**
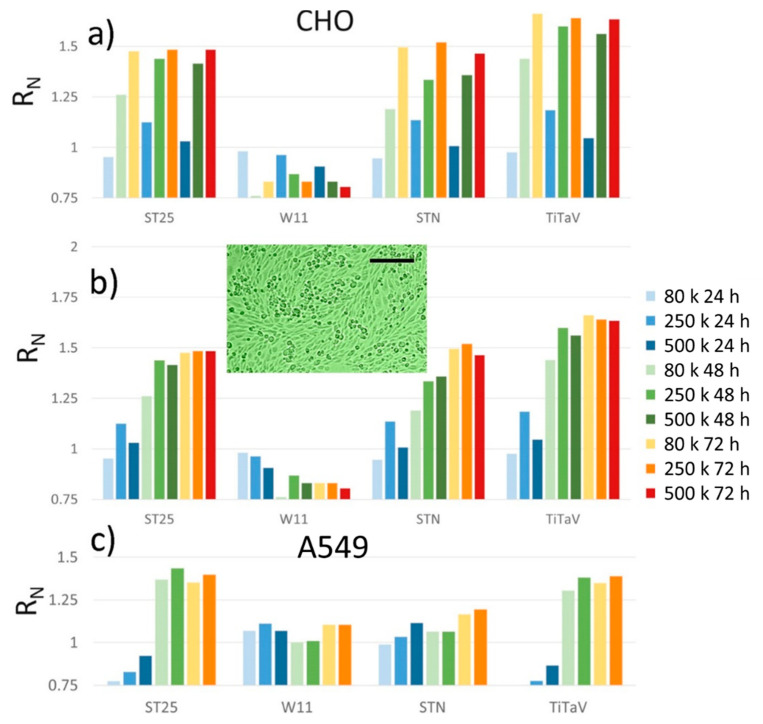
Sensor responses to CHO and A549 cell lines. Cells plated at 80 K (lightest colors), 250 K (light colors), 500 K (medium dark colors), and 1 M (darkest colors) cells/dish, measured after being withdrawn from the incubator at 24 h (blue), 48 h (green) and 72 h (orange); (**a**) CHO cells sorted for the same concentration at different times of incubation; (**b**) CHO cells sorted for the same time of incubation but at different concentrations; the TiTaV response to A549 plated at 80 K cells/dish is not visible because it was smaller than 0.75; (**c**) A549 cells, same color coding as (**b**). Insert, microphotograph of CHO cells seeded at 500 K cells/dish after 48 h of plating; scale bar is 200 µm.

**Figure 13 nanomaterials-12-01111-f013:**
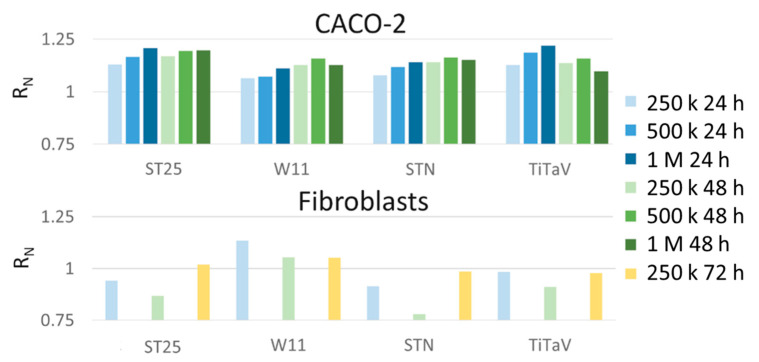
Sensor responses to CACO-2 and fibroblast cell lines. Cells plated at 250 K (light colors), 500 K (medium dark colors), and 1 M (darkest colors) cells/dish, measured after being withdrawn from the incubator at 24 h (blue), 48 h (green) and 72 h (orange); cells sorted for the same concentration at different times of incubation. The sensor responses not visible in the histograms were smaller than 0.75.

**Figure 14 nanomaterials-12-01111-f014:**
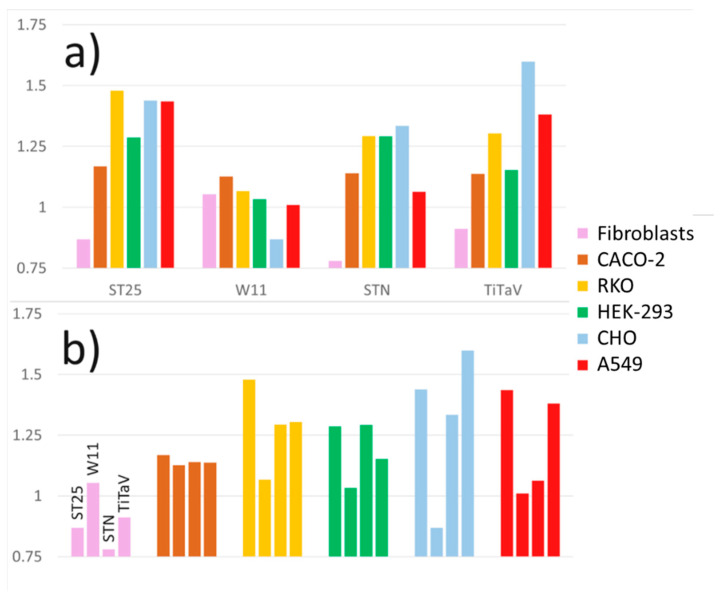
Sensor responses to different cell lines at same plating concentration and withdrawal time from the incubator. (**a**) Responses of the same sensor to different cell types; (**b**) responses of different sensors to the same cell type.

**Table 1 nanomaterials-12-01111-t001:** Hypothetical matrix of the sensor responses to the six culture cells plated at 250 K cells/dish after 48 h of plating.

Cell Type	ST25	STN	TiTaV
Fibroblasts	1.11	0.78	1.17
CACO-2	1.03	1.00	1.14
RKO	1.14	1.29	1.01
HEK-293	1.12	1.12	1.15
CHO	1.08	1.33	1.20
A549	1.35	1.06	1.30

Red numbers represent the smallest response given by one of the three sensors to the VOCs exhaled by a particular cell line.

## Data Availability

Not applicable.

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
