# Peer review of "Chemoresistive Sensors for Cellular Type Discrimination Based on Their Exhalations"

_nanomaterials, 2022, doi:10.3390/nano12071111_

Round 1

Reviewer 1 Report

I am willing to accept the paper in its current form. 

Author Response

We are grateful with this reviewer to finally accept our paper in the present form

Reviewer 2 Report

This paper discussed the detection of VOC of cancer cells, using an Interdigital Electrode, to achieve the goal of building a fingerprint cellular database for cancer cells. Based on the previous comments and responses, the reviewer can understand the idea of the authors, however, it is very difficult for the reviewer to agree with the authors' claim on "it is therefore unnecessary to develop sensor arrays able to single out the molecular composition of cellular VOCs (that would require a very large number of sensors), but to gather instead a limited set of sensors able to univocally identify the cell lines, regardless how much selective is each sensor." If there is no selectivity, it would be hard for the researcher to gain a conclusion. Besides, this device has been patented already, is that still fine for publication? Moreover, there are so many sensors reported using interdigital sensors, which made the reviewer hard to find new contribution of this paper to Nanomaterial field. 

Author Response

First of all, our sensors are chemoresistive ones, therefore their electrical resistance changes with their interaction with the environmental gases (VOCs); the interdigital transducer consists of two interlocking comb-shaped arrays of metallic electrodes, deposited on the surface of a piezoelectric substrate, to form a periodic structure, able to convert electric signals to surface acoustic waves by generating periodically distributed mechanical forces via piezoelectric effect. Our sensors are selective, but not enough to identify the molecular composition of the VOCs: nevertheless, their selectivity sufficed to identify, as demonstrated in this paper, many different cell culture univocally. Our patented device, but hosting a new array of  chemoresistive sensors, has been used here for the first time to analyze the VOCs emitted by six different cell lines, so the authors do not see any particular problem in publishing the results obtained with these new sensors (employed in a new experimental application), although inserted in a patented device. For instance, the gas chromatograph is patented, but many studies obtained with it are daily  published in prestigious journals.

Reviewer 3 Report

The topic and results of the manuscript are very interesting, I would recommend accepting it in present form. 

Author Response

We are grateful with this reviewer to finally accept our paper in the present form

This manuscript is a resubmission of an earlier submission. The following is a list of the peer review reports and author responses from that submission.

Round 1

Reviewer 1 Report

The authors reported the chemoresistive sensors for cellular type discrimination based on their designed sensor.

Generally, this paper is discussing a sensor, without any relation with Nanomaterials, thus can not fit this journal. Besides, from scientific aspect, the quality of the current version is quite low, such as the unclear advantage of this paper comparing the papers published in this field, as well as the the necessity of this paper. Furthermroe, as a sensor, the basic requirements like selectivity, sensitivity and reproducibility, as well as detection mechanisms are not clearly demonstrated, thus making the reviewer hard to provide a more positive recommendation.

Reviewer 2 Report

The paper presents the nanostructured chemoresistive gas sensor to explore the gaseous exhalations of tumoral, immortalized and healthy cell lines with the aim of distinguishing their VOC patterns. The paper is interesting and explained in detail. It has a potential to be considered for publication, however, the English used in the paper is unacceptable and difficult to read. Therefore, I suggest the author use the English Editing service to enhance the readability of the paper.

  • The paper is written in an adverse English. I suggest the author to use an English Editing service to improve the readability of the paper. In most of the places, the sentence formation is inaccurate. Moreover, the tenses (past, present and future) are wrongly used in the formation of the sentences.
  • Figure 5 is not fully explained. Such as 5(a,b,d,e,g,h).
  • Table 1 caption should be placed above the table.
  • Can author explain more about the sol-gel technique for the synthesis of metal oxides. If possible provide the diagram of the synthesis process.
  • Along with figure 1, it is suggested to provide the photo of the complete measurement setup with proper labels. It will help the reader to visualize the sensing process of the proposed device.
  • Figure 2 should be properly labelled for all the concentration or time of incubation.